# A Monte Carlo simulation study of sample size requirements for the Graded Response Model

**Tatsuya Ikeda** [ID]*

Graduate School of Education, Hyogo University of Teacher Education, Kobe, Hyogo, Japan

* ikeda.tatsuya.clpsy@gmail.com

## Abstract

### Background

The graded response model (GRM) is commonly used in psychometrics to analyze ordinal response data. Despite its growing application in scale development and validation, sample size recommendations—such as those provided by the COSMIN guidelines (e.g., $n \geq 1000$)—are often based on expert consensus rather than empirical validation. Furthermore, the extent to which the number of items ($J$) and the number of response categories ($K$) contribute to parameter estimation accuracy remains insufficiently explored.

### Methods

We conducted a Monte Carlo simulation to examine how three design conditions—sample size ($n = 500$–$1500$), number of items ($J = 5$–$50$), and a number of response categories ($K = 4$–$7$)—influence the estimation accuracy of the latent trait parameter ($\theta$) and the item discrimination parameter ($a$) under the GRM. For each condition, we generated a large population dataset based on predefined distributions for $\theta$, $a$, and $b$, and then randomly drew samples ($n$) for estimation. The GRM was fitted using the EM algorithm. Estimation accuracy was evaluated using root mean squared error (RMSE), FPC-corrected RMSE, and Pearson's correlation coefficient between true and estimated $\theta$ values.

### Results

The RMSE of the discrimination parameter $a$ decreased with increasing sample size ($n$) and number of items ($J$), while the effect of $K$ was negligible. In contrast, the RMSE of $\theta$ was primarily influenced by $J$, with only minor effects from $n$ and $K$. Notably, the Pearson correlation between true and estimated $\theta$ values consistently exceeded $r = .98$ across all conditions, suggesting high ordinal fidelity even with small samples. Increasing $J$ beyond approximately 30 yielded diminishing returns in RMSE reduction.

**Data availability statement:** All data underlying the findings of this study are available from the Open Science Framework at the following viwe-only URL: https://osf.io/yw9b7/?view_only=a2a8cb22f1224a2b98f13c69621ac6cd. The data will be made fully public upon publication. Please see the accompanying README file in the repository for variable definitions and figure correspondence.

**Funding:** This work was supported by the Japan Society for the Promotion of Science (JSPS) KAKENHI [Grant Number JP25K14351] awarded to TI. The funder had no role in study design, data collection and analysis, decision to publish, or preparation of the manuscript. URL: https://www.jsps.go.jp/english/.

**Competing interests:** The authors have declared that no competing interests exist.

## Conclusions

Our findings suggest that sample size recommendations for GRM should be flexibly tailored to the measurement goal. For accurate estimation of $\theta$, a sufficiently large number of items (e.g., $J \geq 30$) can compensate for smaller sample sizes ($n \approx 500$), whereas precise estimation of $a$ requires larger samples ($n \geq 1000$). The impact of increasing $K$ was limited, indicating that additional response categories may not always enhance parameter recovery. These results provide empirically grounded guidance to support efficient and purpose-specific measurement designs in GRM applications.

## Introduction

Item Response Theory (IRT) has been widely used across various fields, including education, psychology, healthcare, and industry, as a theoretical framework for test construction and ability measurement. One of the distinctive features of IRT is its ability to estimate both the examinee's latent trait ($\theta$) and item parameters (discrimination and difficulty) simultaneously. This enables item parameter estimates to remain independent of the sampled population and ensures fair comparisons across examinees. Such properties are particularly advantageous for constructing shortened or translated versions of psychological scales. Because many psychological instruments use Likert-type scales with three or more response categories, models that extend IRT from binary to polytomous data are frequently applied. These include the Graded Response Model (GRM) by Samejima [1,2] and the Generalized Partial Credit Model (GPCM) by Masters [3] and Muraki [4,5]. Among these, GRM is especially versatile for items with ordinal responses and is widely used to estimate latent traits and item parameters in psychological research (e.g., Adler et al. [6]; Ikeda and Urano [7]; Wang, Zhang, and Xin [8]; Zhu et al. [9]).

With the increasing use of IRT in empirical studies, there is growing demand for measurement designs that ensure sufficient estimation accuracy of latent traits and item parameters. The COSMIN guidelines [10] recommend a sample size of $n = 1000$ for scale development using IRT models. However, such recommendations are primarily based on expert consensus rather than empirical evidence, and it remains unclear whether a fixed sample size is appropriate across different design conditions. Specifically, as item parameters remain constant, increasing the number of items should increase the information available from the test as a whole, while increasing the number of response categories should theoretically enhance the information obtained from each item. Accordingly, sample size, number of items, and number of response categories are all expected to influence estimation accuracy. This study focuses in particular on the impact of including the number of response categories as a design factor.

Previous research has explored the relationship between design conditions and parameter recovery. For example, Edelen and Reeve [11] emphasized the need for sufficient responses in each category of each item and recommended increasing

sample size as the number of categories grows. Jiang, Wang, and Weiss examined the effects of sample size, number of items, and number of dimensions on parameter estimation and found the largest reduction in bias when the sample size increased from 500 to 1000. Dai and colleagues, incorporating missing data into their analyses of GRM and GPCM, concluded that a minimum of five items and a sample size of at least 500 were necessary when missingness exceeded 20%. These studies collectively underscore the importance of designing sample size requirements with consideration of item- and scale-level characteristics.

A systematic investigation of required sample sizes under different conditions is essential because IRT parameter recovery is strongly influenced by various measurement design factors. In particular, when researchers apply IRT to existing psychological instruments, the only design parameter they can typically manipulate is the sample size. Thus, many studies have focused on determining appropriate sample sizes. Doostfatemeh, Ayatollah, and Jafari [12] highlighted that sample size planning under IRT differs from classical test theory (CTT), and applying CTT-based planning to IRT analyses could yield inaccurate conclusions. As noted by Schroeders et al. [13], sample size decisions should account for the underlying model and test design, and simulation-based approaches are essential for this purpose—though, as they point out, the true item and trait distributions are rarely known, making such simulations inherently challenging.

The required sample size also differs because psychological scales vary in number of items and response categories depending on the construct being measured. Svetina Valdivia and Dai [14] reported that for latent trait and discrimination parameter recovery in polytomous IRT models, a sample size of at least 250 and a higher number of items led to more accurate estimates. However, the effect of response category number was less conclusive. The interaction between sample size and number of items has also been suggested as important. Şahin and Anıl [15] conducted a large-scale simulation varying these two factors and found that acceptable parameter recovery required larger samples when item counts were low, and more items when sample sizes were small. They noted a trade-off between these two factors: more items can compensate for smaller samples, and vice versa, due to their combined contribution to test information.

While a considerable body of research has addressed IRT sample size requirements, few studies have systematically investigated how all three design variables—sample size, number of items, and number of response categories—jointly influence parameter recovery. This gap is particularly relevant for GRM, which assumes ordinal response categories. Additionally, although most prior studies have evaluated estimation accuracy using absolute error metrics such as RMSE, few have included intuitive measures such as the Pearson correlation between true and estimated $\theta$ to assess ordinal accuracy. There is a growing need for empirical guidance on sample size design tailored to varying test information levels and for intuitive metrics to evaluate estimation accuracy.

This study aims to clarify how sample size ($n$), number of items ($J$), and number of response categories ($K$) affect the estimation accuracy of latent trait ($\theta$) and item discrimination ($a$) parameters under the GRM. Focusing on GRM, which inherently assumes variation in $K$, we conduct a comprehensive simulation to evaluate estimation accuracy across a wide range of design conditions, thereby providing a theoretical basis for IRT-based measurement planning.

This study makes two methodological contributions to IRT-based research. First, it is one of the few to systematically examine the combined influence of sample size, number of items, and number of response categories—factors known to affect parameter recovery. While previous studies (e.g., Jiang et al. [16]; Dai et al. [17]) fixed $K$, this study explicitly varies it, enabling more comprehensive insight into the effect of response scale granularity on GRM estimation. This approach extends existing findings and supports guideline refinement.

Second, the study introduces the use of Pearson correlation to assess the ordinal fidelity of $\theta$ estimation. In addition to RMSE, we evaluate average correlations between true and estimated $\theta$ values, allowing for more intuitive interpretation of estimation accuracy from a psychometric perspective. This dual evaluation approach provides differentiated insights for person- and item-level estimation.

## Materials and methods

### Data generation

We conducted a simulation study based on the Graded Response Model (GRM) proposed by Samejima [1,2]. The manipulated design conditions included the number of response categories ($K = 4 - 7$), number of items ($J = 5 - 50$, in increments of 5), and sample size ($n = 500 - 1500$, in increments of 100). All combinations of these factors were considered, and 100 replications ($R = 100$) were generated for each condition.

In each simulation, the population size was set to $N_{pop} = \lceil n/0.001 \rceil$, and a sample of $n$ individuals was randomly drawn from this population. This ensured that all samples shared a common parameter structure $(\theta, a, b)$.

The parameters were generated from the following distributions. The latent trait $\theta_i$ was generated independently for all $N_{pop}$ individuals from the standard normal distribution $\mathcal{N}(0, 1)$. Following previous GRM simulation studies [12,17], the discrimination parameters $a_j$ were drawn from a uniform distribution $\mathcal{U}(0.5, 2.5)$ for each of the $J$ items and the category location parameters $b_{jk}$ were drawn from $\mathcal{U}(-2, 2)$, generating $K-1$ thresholds per item, sorted in ascending order.

The category thresholds $d_{jk}$ were computed using the following equation:

$$d_{jk} = -Da_j b_{jk} \tag{1}$$

where the scaling constant was set to $D = 1.701$. Based on this structure, category responses were generated using the GRM response probabilities. The random seed was set to 123.

### Sampling, estimation, and model considerations

For each fixed population, a sample of size $n$ was randomly drawn, and a unidimensional GRM was fitted to the observed response data. Parameter estimation was performed using the expectation-maximization (EM) algorithm, which is widely employed in applied IRT analyses.

It is important to note that even if the entire population (i.e., $n = N_{pop}$) were observed, the latent trait $\theta$ is not an observable variable, and therefore the difference between estimated and true values is not necessarily zero. Consequently, RMSE is theoretically never zero and always contains some degree of estimation error. This reflects a structural property of latent variable models: even in a full census, uncertainty in estimation remains.

### Parameter estimation

For each generated response dataset, a unidimensional GRM was fitted to estimate the item parameters and latent traits. The item discrimination parameters $\hat{a}_j$ were extracted from the fitted model. The latent traits $\hat{\theta}_i$ were estimated using Expected A Posteriori (EAP) estimation based on the posterior distribution.

### Evaluation metrics

Estimation accuracy was evaluated from multiple perspectives, including the magnitude of errors, standardized errors, corrected errors, and ordinal consistency. Separate indices were calculated for item parameters ($a$) and person parameters ($\theta$).

**Evaluation of discrimination parameter $a$.** The root mean squared error (RMSE) reflects the magnitude of the difference between true and estimated values, and lower values indicate higher accuracy. The RMSE for the estimated discrimination parameters $\hat{a}_j$ was defined as:

$$\text{RMSE}_a = \sqrt{\frac{1}{J} \sum_{j=1}^{J} (\hat{a}_j - a_j)^2} \tag{2}$$

To address the potential downward bias of RMSE when the number of items is small, we applied a finite population correction (FPC), defined as:

$$FPC_a = \sqrt{\frac{J}{J-1}} \tag{3}$$

The FPC-corrected RMSE was then computed as:

$$RMSE_a^{FPC} = RMSE_a \cdot FPC_a \tag{4}$$

To improve comparability across conditions, we also defined a standardized RMSE based on the observed range of $\hat{a}_j$ values:

$$RMSE_a^* = \frac{RMSE_a}{\max(\hat{a}_j) - \min(\hat{a}_j)} \tag{5}$$

The FPC-corrected version of the standardized RMSE was:

$$RMSE_a^{*FPC} = RMSE_a^* \cdot FPC_a \tag{6}$$

**Evaluation of Latent Trait $\theta$.** The RMSE for the latent trait estimates $\hat{\theta}_i$ was defined as:

$$RMSE_\theta = \sqrt{\frac{1}{n}\sum_{i=1}^{n}(\hat{\theta}_i - \theta_i)^2} \tag{7}$$

To adjust for potential bias due to high sampling ratios, we introduced the following FPC based on the sample size $n$ and population size $N_{pop}$:

$$FPC_\theta = \sqrt{1 - \frac{n}{N_{pop}}} \tag{8}$$

The corrected RMSE was then calculated as:

$$RMSE_\theta^{FPC} = RMSE_\theta \cdot FPC_\theta \tag{9}$$

To assess ordinal consistency between $\hat{\theta}$ and $\theta$, we also computed the Pearson correlation coefficient $r$, defined as:

$$r = \frac{\sum_{i=1}^{n}\left(\theta_i - \bar{\theta}\right)\left(\hat{\theta}_i - \bar{\hat{\theta}}\right)}{\sqrt{\sum_{i=1}^{n}\left(\theta_i - \bar{\theta}\right)^2}\sqrt{\sum_{i=1}^{n}\left(\hat{\theta}_i - \bar{\hat{\theta}}\right)^2}} \tag{10}$$

To obtain a more stable estimate of the mean correlation across replications, we applied Fisher's $z$-transformation:

$$\bar{z} = \frac{1}{R}\sum_{i=1}^{R} \tanh^{-1}(r_i) \tag{11}$$

$$\bar{r} = \tanh(\bar{z}) \tag{12}$$

## Software

All simulations and analyses were conducted using R (Version 4.4.3) [18]. The `mirt` package [19,20] was used for GRM data generation and parameter estimation. To enable efficient batch processing and parallel computation on large population data-sets, we employed the `future` [21] and `future.apply` [22] packages. Some parts of the parameter generation process were implemented in C++ using the `Rcpp` package [23]. All code was executed in an RStudio environment running on macOS.

## Evaluation criteria

In this study, we used RMSE = 0.30 as a *convenient reference point* to indicate "sufficient accuracy," rather than as a universal benchmark. This value corresponds to ±1 SD under a standard normal assumption $\mathcal{N}(0, 1)$, meaning that approximately 68% of estimation errors fall within ±0.30:

$$P(|\hat{\theta}_i - \theta_i| < 0.30) = P\left(-1 < \frac{\hat{\theta}_i - \theta_i}{0.30} < 1\right) \approx 0.6827.$$

(13)

Changing the threshold naturally alters which conditions are classified as "sufficient," particularly the **required number of items**, but it does not materially affect the overall trends regarding **sample size** or **number of categories**. Researchers who prefer stricter or more lenient criteria may adjust the RMSE threshold (e.g., < 0.30 or > 0.30) according to their desired confidence level, and our results can be interpreted accordingly.

To supplement this, we also evaluated the consistency between true and estimated $\theta$ using the average Pearson correlation coefficient. This coefficient indicates how accurately the individual-level variance in the latent trait is captured by the estimates. Based on Cohen's guidelines [24], we interpreted values above 0.70 as "high agreement," 0.50–0.70 as "moderate agreement," and below 0.50 as "low agreement." This interpretation also applies to the average correlation coefficients obtained through Fisher's *z*-transformation. Therefore, in this study, we considered a mean correlation above 0.70 as indicative of strong ordinal agreement between estimated and true values.

For the discrimination parameter *a*, which was generated from a uniform distribution and did not assume normality, absolute benchmarks like those used for $\theta$ were not applicable. Instead, we focused on the relative change in $\text{RMSE}_a$ across sample sizes. Since estimation inevitably involves error, RMSE will never reach zero. However, it is expected to asymptotically approach a lower bound as sample size increases. This plateau indicates the point at which additional data yield diminishing returns in estimation precision. We therefore used the point of RMSE stabilization as a practical guideline for determining adequate sample size for *a* estimation.

To further reassure readers about the stability of our simulation results, we note that they relied on independent replications with deterministic EM estimation rather than MCMC sampling; therefore, traditional convergence diagnostics (e.g., Gelman–Rubin statistics) were not applicable. Instead, we assessed stability by inspecting RMSE values across increasing numbers of items *J*. The observed plateau in RMSE indicates that additional items yield diminishing improvements in estimation accuracy, serving as a practical confirmation of convergence in this simulation context.

## Results

### Estimation accuracy of the discrimination parameter *a*

**Effects of sample size, number of items, and number of categories on non-standardized RMSE.**  The RMSE for the discrimination parameter *a* decreased with increasing sample size (*n*) and number of items (*J*). In particular, as *n* increased, RMSE consistently declined (Fig 1). For example, under the condition of *J* = 10, *K* = 5, the mean RMSE decreased from approximately 0.243 at *n* = 500 to about 0.143 at *n* = 1500, indicating that increasing the sample size improves the estimation accuracy of *a*. This trend was also observed across other *J* and *K* conditions.

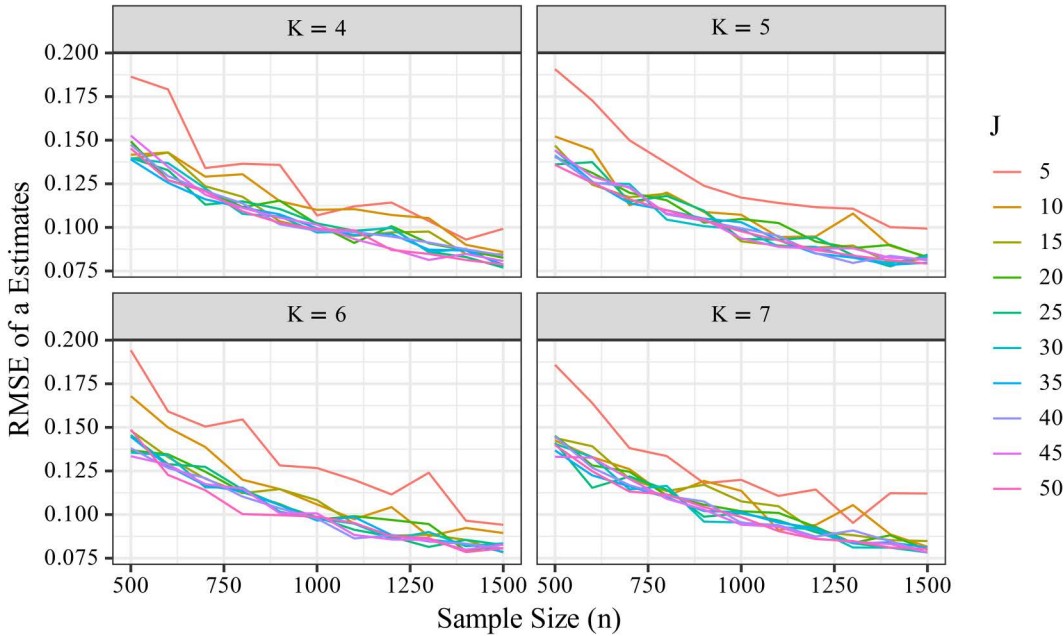

**Fig 1. Changes in non-standardized RMSE of the discrimination parameter *a* with increasing sample size (*n*).** The RMSE consistently decreased as *n* increased, indicating that larger sample sizes contributed to improved estimation accuracy of the discrimination parameter.

To visualize this trend by item count and examine the variability of the RMSE distribution, a boxplot was constructed (Fig 2). As shown in Fig 2, increasing *n* consistently reduced RMSE, demonstrating that larger sample sizes contribute to more accurate estimation of *a*. In contrast, changes in *K* had minimal effect on RMSE. The RMSE decreased similarly with increasing *n*, regardless of *K*, indicating that the additional information provided by more response categories contributed little to the accuracy of *a*.

**Effects on standardized FPC-corrected RMSE.** The standardized FPC-corrected RMSE for the discrimination parameter *a* gradually decreased with increasing number of items. Fig 3 illustrates how this metric varied with sample size (*n*) and number of items (*J*), and Fig 4 shows that RMSE declined gradually as *J* increased.

To more clearly visualize the influence of *J*, we averaged RMSE values across different *K* levels under the same *J* and *n* conditions (Fig 5). The results showed a gradual decline in RMSE as *J* increased, but the effect was modest compared to the influence of *n*. Notably, under conditions with small *n* (e.g., *n* = 500), increasing *J* did not substantially reduce RMSE, suggesting that increasing the number of items alone is insufficient for improving the precision of *a*. In addition, the contribution of more items to estimation accuracy was most notable between *J* = 10 and *J* = 20, with diminishing returns observed beyond *J* = 30.

Fig 6 shows changes in $\text{RMSE}_a^{*\text{FPC}}$ across combinations of *J* and *n*. Improvements in estimation accuracy due to increasing *J* were particularly pronounced when *J* < 30. The effect of *n* was relatively consistent and robust. However, the improvement in accuracy diminished as *J* increased, with no substantial gains beyond *J* ≥ 30.

### Estimation accuracy of the latent trait parameter *θ*

**Effects on FPC-corrected RMSE.** To evaluate the estimation accuracy of *θ*, we computed RMSE and FPC-corrected RMSE and examined their changes across conditions of *J*, *K*, and *n*. Fig 7 shows that RMSE for *θ* did not decrease substantially as *n* increased. For instance, comparing conditions with *n* = 500 and *n* = 1500, the mean RMSE differed only

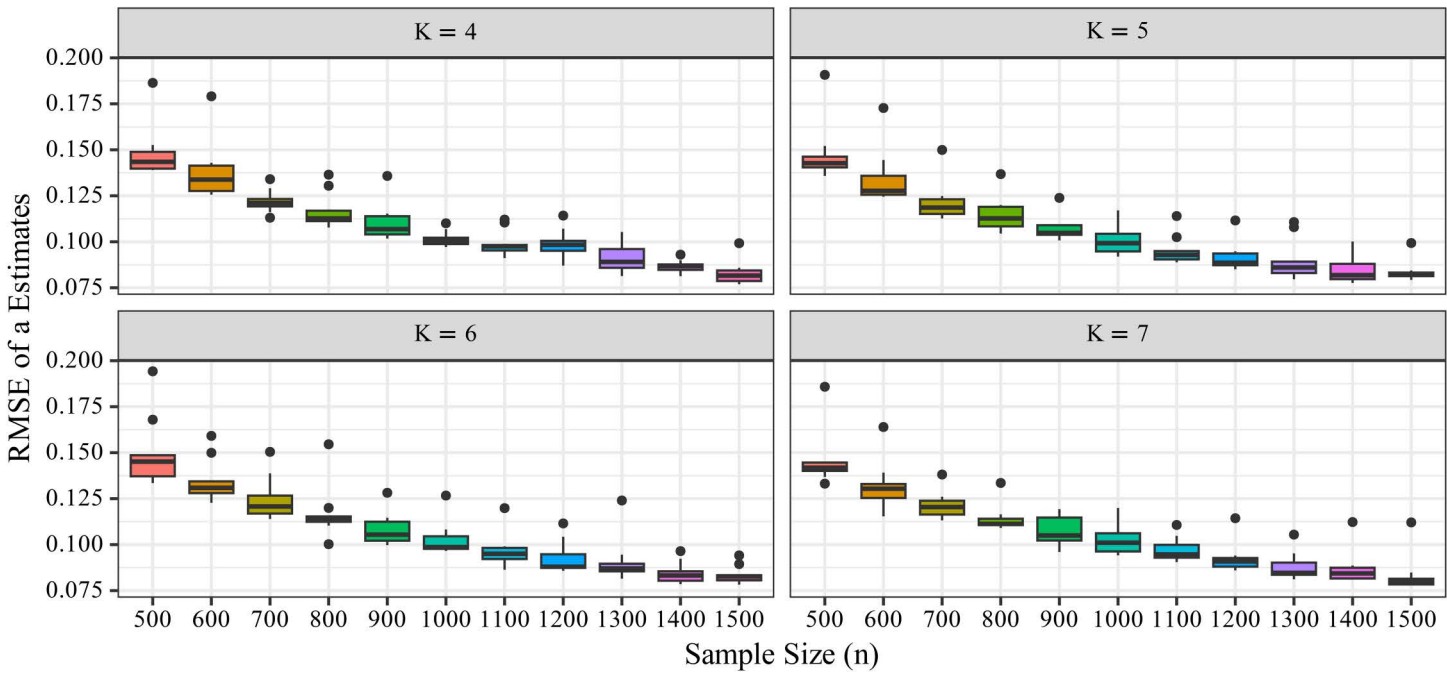

**Fig 2. Changes in RMSE$_a$ with sample size ($n$).** The boxplots show the distribution of estimation errors under each condition. The parameter $a$ represents the slope parameter in the Graded Response Model (GRM), and RMSE$_a$ indicates the estimation accuracy for $a$. The x-axis represents the sample size ($n$), and the y-axis represents the RMSE of the estimated slope parameters. The number of response categories ($K$) is indicated by panel or grouping as appropriate.

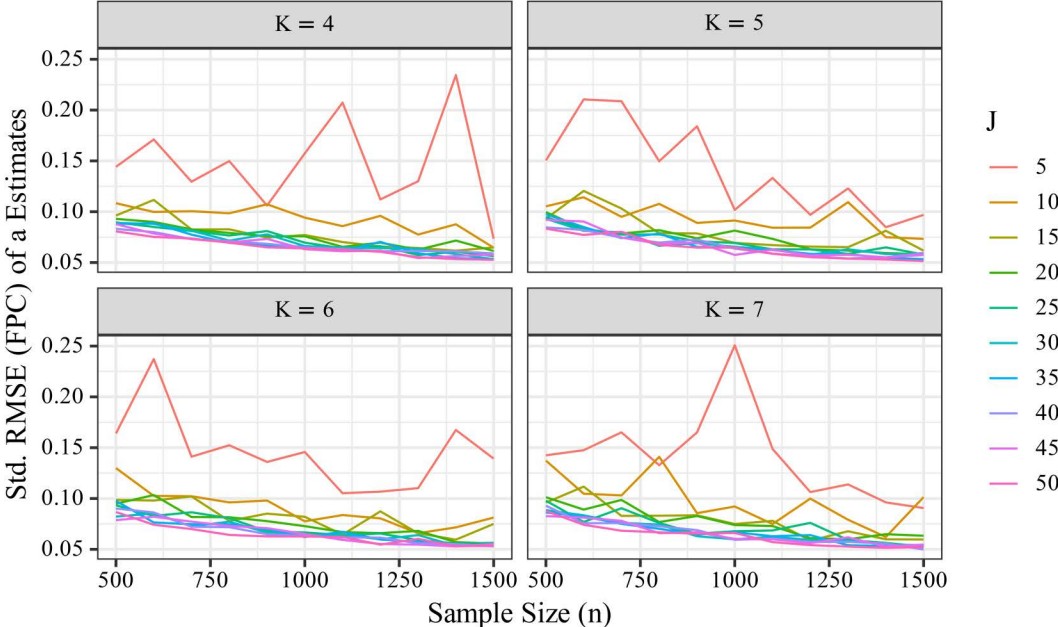

**Fig 3. Effects of sample size ($n$) and number of items ($J$) on the RMSE$_a^{*FPC}$.** The RMSE$_a^{*FPC}$ gradually decreased with increasing $J$, although the effect was modest compared to the influence of $n$.

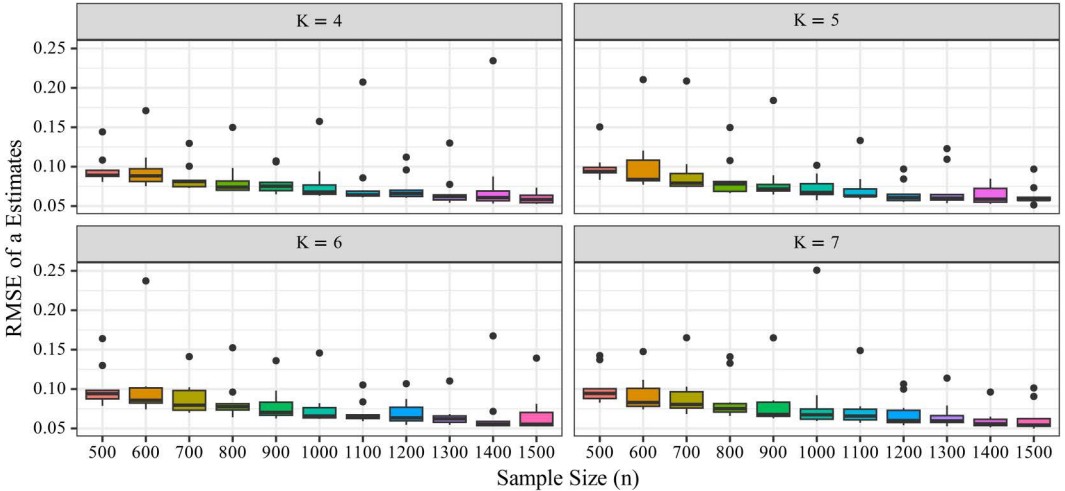

**Fig 4. Transition of RMSE$_a^{*FPC}$ with sample size ($n$).** Boxplots show the distribution of standardized and FPC-corrected RMSE values for the discrimination parameter $a$. The x-axis represents sample size ($n$), and the y-axis represents RMSE. The number of response categories ($K$) is indicated by panel or grouping as appropriate.

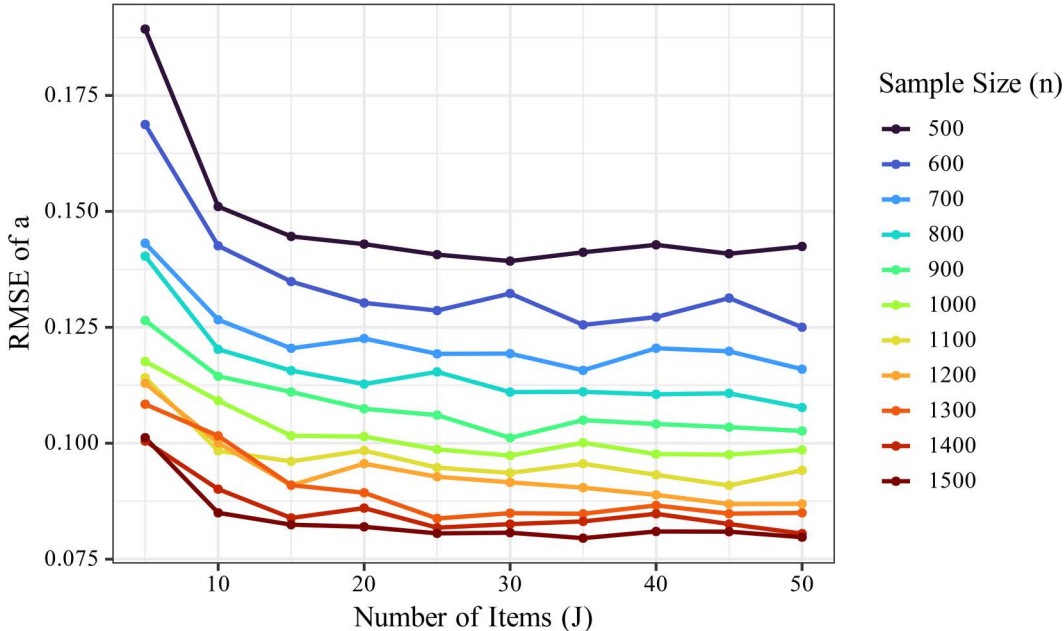

**Fig 5. Changes in RMSE$_a$ across the number of items ($J$).** The line plot shows how the RMSE of the discrimination parameter ($a$) changes with increasing item count ($J$). The x-axis represents the number of items ($J$), and the y-axis represents RMSE$_a$.

slightly. In contrast, Fig 8 shows a marked reduction in RMSE with increasing $J$, particularly when comparing $J=5$ and $J=50$. The influence of $K$ on RMSE was minimal in comparison to that of $J$, although a slight improvement was observed.

We also compared uncorrected RMSE values for $\theta$ (Supplementary S1 and S2 Figs) with the FPC-corrected results shown in Figs 7 and 8. The differences between corrected and uncorrected values were negligible—visible changes were

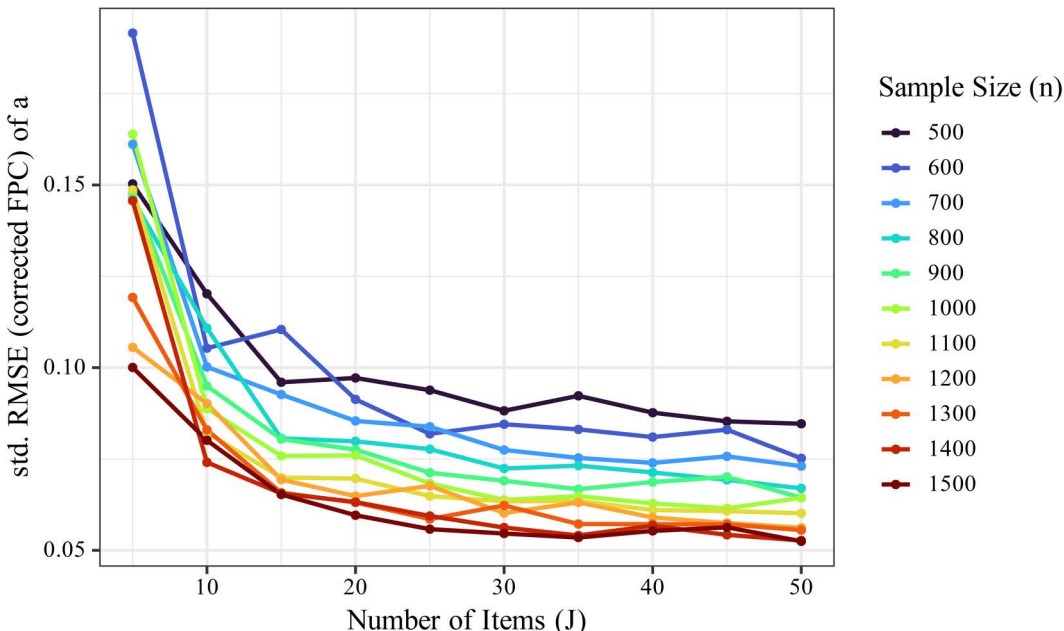

**Fig 6. Changes in RMSE$_a^{*FPC}$ across the number of items (*J*).** The line plot shows how the RMSE$_a^{*FPC}$ changes with increasing number of items (*J*). The RMSE values were corrected using finite population correction (FPC). The x-axis represents the number of items (*J*), and the y-axis represents RMSE$_a^{*FPC}$.

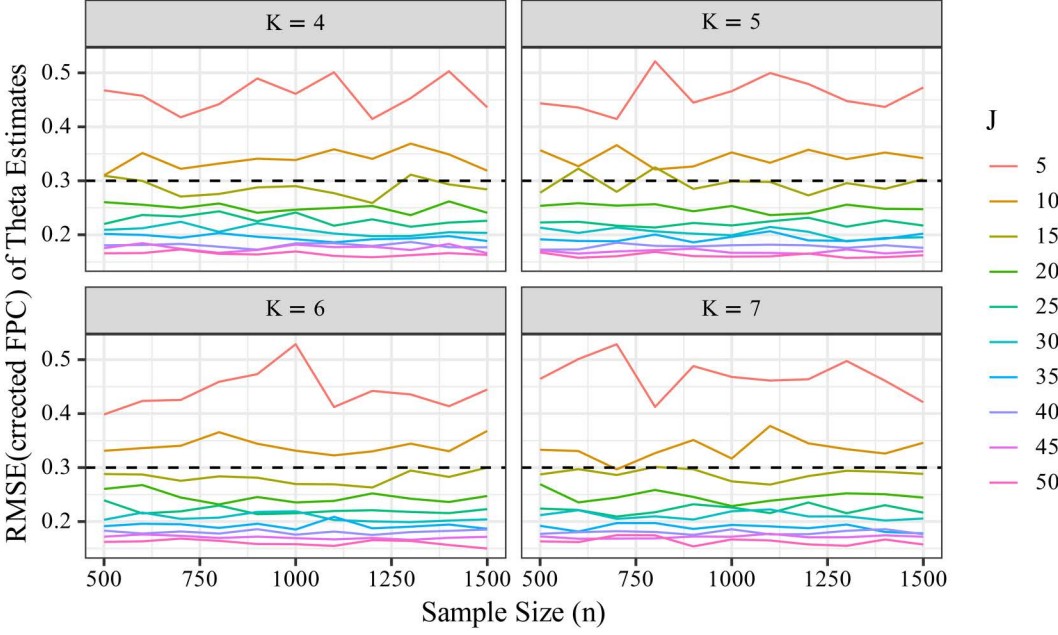

**Fig 7. Changes in RMSE$_\theta^{FPC}$ by sample size (*n*), number of items (*J*), and number of response categories (*K*).** The line graph illustrates how the FPC-corrected RMSE of the latent trait parameter ($\theta$) varies across different levels of *n*, *J*, and *K*.

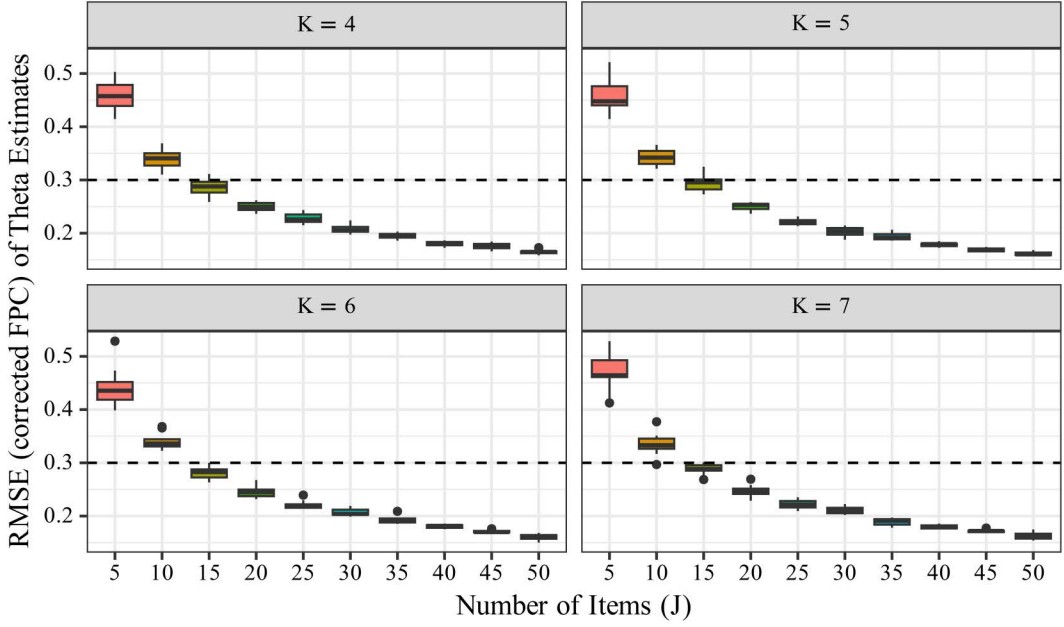

**Fig 8. Changes in RMSE$_\theta^{FPC}$ by number of items (J).** The boxplots show the distribution of RMSE$_\theta^{FPC}$ under each condition. The x-axis represents the number of items (J), and the y-axis represents the FPC-corrected RMSE of the estimated latent trait ($\theta$).

absent in the figures, and numerical differences occurred only at the third decimal place. This confirms that the FPC correction has virtually no impact on the evaluation of $\theta$ estimation accuracy.

**Ordinal consistency between true and estimated $\theta$.** To examine the relationship between $\theta$ and $\hat{\theta}$, we computed the average Pearson correlation coefficient and visualized the results across all conditions (Fig 9). Similar to the RMSE findings, Fig 9 shows that the mean correlation remained stable regardless of sample size. For example, under J=50, the mean Fisher-transformed correlation ranged from.985 to.989, indicating minimal fluctuation and high consistency regardless of n.

Unlike RMSE, the correlation between $\theta$ and $\hat{\theta}$ exceeded the benchmark of r>.70 across all levels of J. This result suggests that item count does not meaningfully affect the ordinal agreement between true and estimated values of $\theta$, a point that is further discussed in the Discussion section.

## Discussion

This study aimed to examine how three measurement design conditions in IRT—sample size (n), number of items (J), and number of response categories (K)—affect the estimation accuracy of latent traits ($\theta$) and item discrimination parameters (a) in the Graded Response Model (GRM). The results indicated that (1) RMSE for a improved with increasing n and J, while the effect of K was minimal; (2) RMSE for $\theta$ decreased substantially with increasing J, but showed limited sensitivity to n and K; and (3) the correlation coefficient between $\theta$ and $\hat{\theta}$ consistently exceeded.98 across all conditions, demonstrating high ordinal consistency.

The improvement in a estimation accuracy with larger n and J can be attributed to two factors: (i) a larger n enables more responses per item, enhancing the precision of item-level slope estimates, and (ii) increasing J raises the total test information, thereby supporting more accurate estimation. In contrast, increasing K did not enhance accuracy likely because estimating category boundaries requires higher-order gradient information. With more categories, the density of information per category decreases, diluting the effectiveness of additional categories for estimating a.

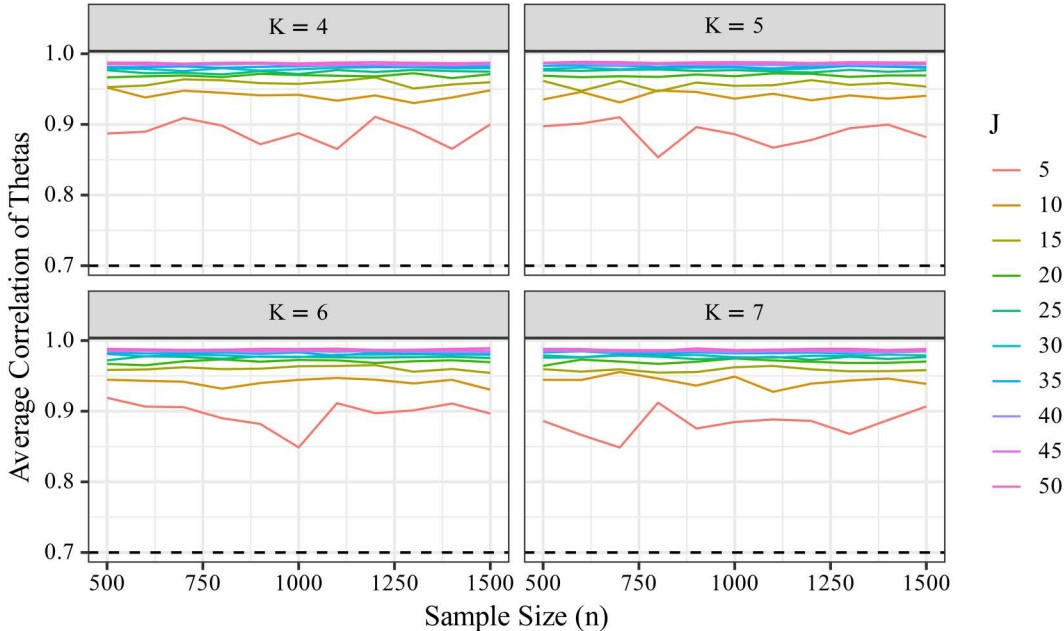

**Fig 9. Changes in the correlation coefficient ($\bar{r}$) between $\theta$ and $\hat{\theta}$ by sample size ($n$).** The line plot shows how the Fisher-transformed correlation coefficient ($\bar{r}$) between true and estimated values of $\theta$ varies with increasing sample size ($n$). The number of items ($J$) is indicated as appropriate.

The improvement of $\theta$ estimation primarily through increases in $J$, with only limited contributions from $n$ and $K$, aligns with IRT theory: the precision of $\theta$ estimates depends largely on total test information rather than sample size. The consistently high correlations between $\theta$ and $\hat{\theta}$ suggest that individual differences in true scores were stably captured by the estimates, indicating that ordinal ranking can be preserved even with smaller samples.

Prior studies have emphasized the trade-off between sample size and item count. For example, Şahin and Anıl [15] showed that smaller item sets require larger samples for adequate parameter recovery. Our study not only confirmed this, but also provided the first quantitative evidence that estimation accuracy for both $\theta$ and $a$ saturates around $J \approx 30$, beyond which further increases in $J$ yield diminishing returns. This finding underscores that unlimited item expansion may not be cost-effective, highlighting the need for rational allocation of measurement resources.

Regarding the number of response categories, Edelen and Reeve [11] noted that large $n$ is required to ensure sufficient responses in each category, while Svetina et al. [14] found that increasing $K$ does not necessarily improve accuracy. Our results support this trend, showing limited gains from higher $K$ for both $\theta$ and $a$. Although more categories theoretically increase information, skewed category usage in practice can cancel out these gains. Therefore, increasing $K$ is not always desirable in applied contexts.

Importantly, our study evaluated $\theta$ estimation not only by RMSE but also by correlation with $\hat{\theta}$, introducing an ordinal perspective on estimation fidelity. This approach aligns with the recommendation of Schroeders et al. [13] to adopt more intuitive accuracy indices. Our findings suggest that correlation, which consistently hovered around.98, may be a more practical indicator than RMSE when ordinal ranking is the primary objective.

Furthermore, our findings invite a reassessment of the COSMIN guideline recommending $N = 1000$. For $\theta$ estimation [10], sufficient accuracy can be achieved with $n \approx 500$ if $J \approx 30$, while $a$ estimation benefits from larger $n$. Thus, the required sample size differs depending on whether the primary goal is estimating person or item parameters. Our results provide a theoretical basis for tailoring sample size recommendations to specific measurement objectives and call for more flexible application of existing guidelines.

This study has three limitations. First, we restricted our analysis to unidimensional GRM, whereas many psychological scales are multidimensional. In multidimensional models, more parameters must be estimated, and the saturation points we report (e.g., for RMSE) may underestimate the necessary sample size per dimension.

Second, the distributions of $a$ and $b$ were fixed as uniform. In practice, $a$ often clusters below 1.0, and $b$ values tend to concentrate around moderate difficulty levels. Such skewed distributions may distort the test information function and reduce the generalizability of our precision metrics.

Third, we did not incorporate missing data or local item dependence. The simulation assumed complete, independent responses, whereas real-world data often include missing values or local dependencies. These factors may inflate estimation error beyond the theoretical values we report. Therefore, our sample size guidelines should be viewed as minimum requirements under ideal conditions.

Future studies should extend this work to multidimensional GRM or GPCM models, explore the effects of parameter distribution skewness, missingness, and local dependence, and develop estimation-agnostic guidelines through comparisons with Bayesian or MIRT estimation. Additionally, optimizing design algorithms based on test information functions could enable dynamic configuration of $n$, $J$, and $K$.

Design decisions should reflect the intended purpose of the scale. If accurate $\theta$ estimation is the goal, $J \approx 30$ may suffice with $n \approx 500$. In contrast, if precise $a$ estimation or item characteristic curve analysis is prioritized, $n \geq 1000$–1500 and $J \geq 20$ are recommended. Since the benefits of increasing $K$ are limited, 4–5 categories may be more practical when considering respondent burden. These findings complement existing guidelines such as COSMIN and support the development of flexible, goal-specific measurement design strategies.

## Conclusion

This study systematically investigated how three design conditions—sample size ($n$), number of items ($J$), and number of response categories ($K$)—influence the estimation accuracy of latent traits ($\theta$) and item discrimination parameters ($a$) under the Graded Response Model (GRM). The findings demonstrated that the accuracy of $\theta$ estimates primarily improved with an increasing number of items, while the impact of sample size and number of categories was limited. In contrast, the accuracy of $a$ estimates improved with both larger sample sizes and more items, whereas increasing the number of response categories had minimal effect.

The estimated values of $\hat{\theta}$ consistently maintained a high correlation with the true $\theta$ values ($r > .98$), indicating strong ordinal agreement even under small sample conditions. In addition, RMSE values for both $\theta$ and $a$ tended to plateau around $J \approx 30$, suggesting that increasing the number of items beyond this point yields diminishing returns in estimation precision.

These results provide empirical evidence for the rational optimization of measurement design in IRT applications, particularly when choosing between prioritizing person-level ($\theta$) versus item-level ($a$) accuracy. Importantly, the study suggests that the sample size recommendation of $N = 1000$, as advocated by the COSMIN guidelines, should not be applied uniformly. Instead, it should be tailored to the primary measurement objective. For instance, if the focus is on estimating $\theta$, a sample size of $n \approx 500$ may be sufficient when $J \approx 30$. However, if the goal is to obtain precise estimates of $a$ or analyze item characteristic curves, a larger sample ($n \geq 1000$–1500) and moderate item count ($J \geq 20$) are recommended. Given the limited benefits of increasing $K$, a practical choice of 4 or 5 categories may be appropriate, especially when considering respondent burden.

Taken together, these findings complement existing guidelines such as COSMIN and support the development of flexible and goal-specific measurement design strategies in psychological and health-related assessment contexts.

## Supporting information

**S1 Fig. Changes in RMSE$_\theta$ by sample size ($n$), number of items ($J$), and number of response categories ($K$).** There is little difference from Fig. 7 using FPC-corrected RMSE.
(PDF)

**S2 Fig. Changes in RMSE$_\theta$ by number of items (*J*).** There is little difference from Fig. 8 using FPC-corrected RMSE. (PDF)

## Author contributions

**Conceptualization:** Tatsuya Ikeda.

**Data curation:** Tatsuya Ikeda.

**Formal analysis:** Tatsuya Ikeda.

**Funding acquisition:** Tatsuya Ikeda.

**Investigation:** Tatsuya Ikeda.

**Methodology:** Tatsuya Ikeda.

**Project administration:** Tatsuya Ikeda.

**Software:** Tatsuya Ikeda.

**Validation:** Tatsuya Ikeda.

**Visualization:** Tatsuya Ikeda.

**Writing – original draft:** Tatsuya Ikeda.

**Writing – review & editing:** Tatsuya Ikeda.

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
