## [Decision Letter · Decision Letter 0]

17 Sep 2025

Dear Dr. Ikeda,

Thank you for submitting your manuscript to PLOS ONE. After careful consideration, we feel that it has merit but does not fully meet PLOS ONE’s publication criteria as it currently stands. Therefore, we invite you to submit a revised version of the manuscript that addresses the points raised during the review process.

plosone@plos.org. . . . A rebuttal letter that responds to each point raised by the academic editor and reviewer(s). You should upload this letter as a separate file labeled 'Response to Reviewers'.A marked-up copy of your manuscript that highlights changes made to the original version. You should upload this as a separate file labeled 'Revised Manuscript with Track Changes'.An unmarked version of your revised paper without tracked changes. You should upload this as a separate file labeled 'Manuscript'.

We look forward to receiving your revised manuscript.

Kind regards,

Mohamed R. Abonazel, Ph.D.

Academic Editor

PLOS ONE

Journal Requirements:

This study was supported by JSPS KAKENHI Grant Number JP25K14351. The authors declare no competing interests.

This work was supported by the Japan Society for the Promotion of Science (JSPS) KAKENHI [Grant Number JP25K14351] awarded to TI. The funder had no role in study design, data collection and analysis, decision to publish, or preparation of the manuscript. URL: https://www.jsps.go.jp/english/

Reviewers' comments:

Reviewer's Responses to Questions

**Comments to the Author**

1. Is the manuscript technically sound, and do the data support the conclusions?

Reviewer #1: Yes

2. Has the statistical analysis been performed appropriately and rigorously?

Reviewer #1: Yes

3. Have the authors made all data underlying the findings in their manuscript fully available?

Reviewer #1: No

4. Is the manuscript presented in an intelligible fashion and written in standard English?

Reviewer #1: Yes

Reviewer #1: Comments to Author:

The current study aimed to test and find an appropriate sample size estimation method to support GRM. The topic is of method importantce, yet there remain issues to improve the work.

1. In the introduction section, academic writing seldom starts with “because” (page 2, line 40 and line 52) as a full sentence, the authors are suggested to ask native English speakers for advice.

2. In the method section, I wonder if the authors could provide justifications for their choice of distribution for the model parameters (page 3, lines 100~105).

3. In the method section, authors use 0.3 as threshold for RMSE to derive a probability of 0.68 for the level of accuracy. I wonder if these settings are supported by previous publications. Also, I wonder if increased threshold levels would make the accuracy level higher and yield better results.

4. In the method section, regarding the line “RMSE reflects the magnitude of the difference between true and estimated…”, I think it should be provided much earlier in the section regarding “Evaluation of Discrimination Parameter a”.

5. Regarding the Monte Carlo simulation methods, I wonder if some sort of convergence diagnostics should be provided to aid readers to understand the improvement in accuracy due to increasing J.

6. Regarding the FPC corrected RMSE, I wonder if the original uncorrected RMSE should also be provided as a comparison to justify the how much these corrections change the estimates.

7. As a methodological paper, I wonder if the authors could further provide some sort of coding scripts and their generated simulated dataset for the sake of replication of their findings.

.

Reviewer #1: No

---

## [Author Response · Author response to Decision Letter 1]

25 Sep 2025

Please see the attached “Response_to_Reviewers.pdf” for the full, formatted responses including mathematical equations.

For convenience, the plain-text version of our responses is also provided below (equations may appear unformatted):

--- Begin copied responses ---

Journal Requirements:

Response:

We have verified our manuscript against the PLOS ONE formatting templates and confirmed that the text structure, headings, references, figures, and tables comply with the journal’s style.

In addition, we have revised all file names to follow the journal’s naming conventions (e.g., fig1.pdf → Fig1.pdf) and renamed our uploaded files as requested:

Revised_Manuscript_with_Track_Changes.pdf (marked-up manuscript)

Manuscript.pdf (clean manuscript)

Response_to_Reviewers.pdf (rebuttal letter)

Figures and supporting information renamed as Fig1.pdf, S1_Fig.pdf, etc.

These changes ensure full compliance with PLOS ONE’s style requirements.

This study was supported by JSPS KAKENHI Grant Number JP25K14351. The authors declare no competing interests.

This work was supported by the Japan Society for the Promotion of Science (JSPS) KAKENHI [Grant Number JP25K14351] awarded to TI. The funder had no role in study design, data collection and analysis, decision to publish, or preparation of the manuscript. URL: https://www.jsps.go.jp/english/

Response:

We have removed funding information from the Acknowledgments section as instructed. The Funding Statement remains unchanged in the submission form, and we have noted this in the cover letter.

Before:

This study was supported by JSPS KAKENHI Grant Number JP25K14351. The authors declare no competing interests. (p. 11, lines 340-341)

After:

The authors declare no competing interests. (p. 11, line 353)

Response;

The reviewers did not recommend any specific publications to cite. Therefore, no changes regarding additional citations were necessary.

Reviewer #1: Comments to Author:

The current study aimed to test and find an appropriate sample size estimation method to support GRM. The topic is of method importantce, yet there remain issues to improve the work.

1. In the introduction section, academic writing seldom starts with “because” (page 2, line 40 and line 52) as a full sentence, the authors are suggested to ask native English speakers for advice.

Response:

We revised the two sentences on page 2 (lines 40 and 52) that began with “Because” to avoid starting a sentence with this conjunction. Other instances of “Because” that were not mentioned by the reviewer were retained unchanged, as they did not occur at the beginning of a sentence or were not flagged as problematic.

Before:

Because IRT parameter recovery is strongly influenced by various measurement design factors, a systematic investigation of required sample sizes under different conditions is essential. (p. 2, lines 40-42)

Because psychological scales vary in number of items and response categories depending on the construct being measured, the required sample size also differs. (p. 2 – 3, lines 52 – 53)

After:

A systematic investigation of required sample sizes under different conditions is essential because IRT parameter recovery is strongly influenced by various measurement design factors. (p. 2, lines 40 – 42)

The required sample size also differs because psychological scales vary in number of items and response categories depending on the construct being measured. (p. 2 – 3, lines 52 – 53)

2. In the method section, I wonder if the authors could provide justifications for their choice of distribution for the model parameters (page 3, lines 100~105).

Response:

We added justifications for the parameter distributions in the Methods section. The latent trait θ_i is now explicitly stated as being drawn from N(0,1), which is standard practice in IRT simulations. Following previous GRM simulation studies (Doostfatemeh et al., 2016; Dai et al., 2021), we generated discrimination parameters a_j from U(0.5,2.5) and category location parameters b_jk from U(-2.0,2.0), sorted in ascending order. This choice avoids bias toward particular parameter regions and ensures broad coverage of the parameter space.

Before:

The discrimination parameters a_j were drawn from a uniform distribution U(0.5,2.5) for each of the J items. The category location parameters b_jk were drawn from U(−2,2), generating K-1 thresholds per item, sorted in ascending order. (p. 4, lines 102 – 104)

After:

Following previous GRM simulation studies [12,17], the discrimination parameters a_jwere drawn from a uniform distribution U(0.5,2.5) for each of the J items and the category location parameters b_jk were drawn from U(−2,2), generating K-1 thresholds per item, sorted in ascending order. (p. 4, lines 102 – 105)

3. In the method section, authors use 0.3 as threshold for RMSE to derive a probability of 0.68 for the level of accuracy. I wonder if these settings are supported by previous publications. Also, I wonder if increased threshold levels would make the accuracy level higher and yield better results.

Response:

We appreciate this insightful comment. We have clarified in the Methods section that RMSE = 0.30 is not a universal benchmark but a convenient reference point corresponding to ±1 SD under a standard normal assumption (≈68% coverage). Changing the threshold naturally shifts which conditions are classified as “sufficient,” especially the required number of items, but it does not materially affect the overall trends regarding sample size or number of categories. Researchers wishing to apply stricter or more lenient criteria can adjust the threshold (e.g., <0.30 or >0.30) accordingly, and our results can be interpreted in that flexible manner.

Before:

In this study, we considered an RMSE of less than 0.30 … the estimation accuracy is considered adequate in practical applications. (p. 6, lines 157 – 165)

After:

In this study, we used RMSE = 0.30 as a convenient reference point to indicate … according to their desired confidence level, and our results can be interpreted accordingly. (p. 6, lines 158 - 167)

4. In the method section, regarding the line “RMSE reflects the magnitude of the difference between true and estimated…”, I think it should be provided much earlier in the section regarding “Evaluation of Discrimination Parameter a”.

Response:

We moved the sentence “RMSE reflects the magnitude of the difference between true and estimated values” to the beginning of the “Evaluation of Discrimination Parameter a” subsection so that readers encounter this explanation before the detailed evaluation. This change clarifies the meaning of RMSE at the point where it is first applied.

Before:

The root mean squared error (RMSE) for the estimated discrimination parameters a �_j was defined as: (p.4, lines 131 - 132)

Instead, we focused on the relative change in RMSEa across sample sizes. RMSE reflects the magnitude of the difference between true and estimated values, and lower values indicate higher accuracy. Since estimation inevitably involves error, RMSE will never reach zero. (p. 6, lines 176 – 179)

After:

The root mean squared error (RMSE) reflects the magnitude of the difference between true and estimated values, and lower values indicate higher accuracy. The RMSE for the estimated discrimination parameters a �_j was defined as: (p. 4, lines 131 – 133)

Instead, we focused on the relative change in RMSEa across sample sizes. Since estimation inevitably involves error, RMSE will never reach zero. (p. 6, lines 178 – 179)

5. Regarding the Monte Carlo simulation methods, I wonder if some sort of convergence diagnostics should be provided to aid readers to understand the improvement in accuracy due to increasing J.

Response:

Thank you for this helpful suggestion. Because our simulations used independent replications with deterministic EM estimation rather than MCMC sampling, traditional convergence diagnostics (e.g., Gelman–Rubin statistics) are not applicable. To clarify this point and to reassure readers about the stability of our results, we added the following explanation to the end of the Evaluation Criteria subsection:

“To further reassure readers about the stability of our simulation results, we note that they relied on independent replications with deterministic EM estimation rather than MCMC sampling; therefore, traditional convergence diagnostics (e.g., Gelman–Rubin statistics) were not applicable. Instead, we assessed stability by inspecting RMSE values across increasing numbers of items J. The observed plateau in RMSE indicates that additional items yield diminishing improvements in estimation accuracy, serving as a practical confirmation of convergence in this simulation context.” (p. 6, lines 184 – 190 in Revised Manuscript)

This addition makes explicit why standard diagnostics were not used and shows that the plateau in RMSE across J provides a practical check of stability.

6. Regarding the FPC corrected RMSE, I wonder if the original uncorrected RMSE should also be provided as a comparison to justify the how much these corrections change the estimates.

Response:

Thank you for pointing this out. For the discrimination parameter a, we had already provided uncorrected (Figs. 1, 2, 5) and FPC‐corrected (Figs. 3, 4, 6) RMSE values. To maintain consistency, we computed and now provide the uncorrected RMSE results for the latent trait θ as Supplementary S1 Fig. (corresponding to Fig. 7) and Supplementary S2 Fig. (corresponding to Fig. 8). As expected, the differences between corrected and uncorrected values are negligible—no visible differences are observed in the figures, and numerical changes occur only at the third decimal place. We have added a sentence in the Results section to clarify that FPC correction has virtually no influence on the RMSE for θ.

Before:

(none)

After:

Supporting information (p. 13)

S1 Fig. Changes in RMSE_θ by sample size (n), number of items (J), and number of response categories (K). There is little difference from Fig. 7 using FPC-corrected RMSE.

S2 Fig. Changes in RMSE_θ by number of items (J). There is little difference from Fig. 8 using FPC-corrected RMSE.

7. As a methodological paper, I wonder if the authors could further provide some sort of coding scripts and their generated simulated dataset for the sake of replication of their findings.

Response:

Thank you for highlighting the importance of reproducibility. The simulated datasets used in this study are already publicly available via OSF at the URL provided in the Data Availability statement of the manuscript. Regarding the simulation code, we are currently refining and packaging the scripts into a user-friendly R package to facilitate broader use and long-term maintenance. At this stage, we prefer not to release the intermediate development version to avoid potential confusion or misuse. However, interested researchers may contact the corresponding author to request the current R scripts. Please note that these scripts contain comments written in non-English (Japanese), which may limit their accessibility. We are committed to making the finalized package publicly available upon completion.

Before:

All data and simulation code used in this study are publicly available at Open Science Framework via the following view-only link: https://osf.io/yw9b7/?view_only=a2a8cb22f1224a2b98f13c69621ac6cd. Please see the accompanying README file for details. (p. 11, lines 343 – 346)

After:

All data used in this study are publicly available at Open Science Framework via the following link: https://osf.io/yw9b7/?view_only=a2a8cb22f1224a2b98f13c69621ac6cd. Please see the accompanying README file for details. The simulation code is currently being refined into a user-friendly R package for broader release; researchers who need the intermediate version may contact the corresponding author (note that the code comments are written in non-English [Japanese]) (p.11, lines 355 – 361)

--- End copied responses ---

---

## [Decision Letter · Decision Letter 1]

6 Apr 2026

A Monte Carlo Simulation Study of Sample Size Requirements for the Graded Response Model

PONE-D-25-26261R1

Dear Dr. Tatsuya Ikeda,

We’re pleased to inform you that your manuscript has been judged scientifically suitable for publication and will be formally accepted for publication once it meets all outstanding technical requirements.

Kind regards,

Leander Luiz Klein, Ph.D.

Academic Editor

PLOS One

**Comments to the Author**

Reviewer #2: All comments have been addressed

Reviewer #3: All comments have been addressed

2. Is the manuscript technically sound, and do the data support the conclusions?

Reviewer #2: Yes

Reviewer #3: Yes

3. Has the statistical analysis been performed appropriately and rigorously?

Reviewer #2: Yes

Reviewer #3: Yes

4. Have the authors made all data underlying the findings in their manuscript fully available?

Reviewer #2: Yes

Reviewer #3: Yes

5. Is the manuscript presented in an intelligible fashion and written in standard English?

Reviewer #2: Yes

Reviewer #3: Yes

Reviewer #2: The mansucript is accepted in it succretn form as the authors have revised the manuscript according to the suggestions and corrections suggested by ythe reviewers

Reviewer #3: (No Response)

.

Reviewer #2: No

Reviewer #3: No

---

## [Editor Report · Acceptance letter]

PONE-D-25-26261R1

PLOS One

Dear Dr. Ikeda,

I'm pleased to inform you that your manuscript has been deemed suitable for publication in PLOS One. Congratulations! Your manuscript is now being handed over to our production team.

Kind regards,

on behalf of

Professor Leander Luiz Klein

Academic Editor

PLOS One